# Establishment of Autoreactive CD4^+^CD8^+^ T Cell Hybridomas from Sjögren’s Disease Model, SATB1 Conditional Knockout Mice

**DOI:** 10.3390/ijms27010414

**Published:** 2025-12-30

**Authors:** Shuhei Mashimo, Michitsune Arita, Taku Kuwabara, Taku Naito, Sakurako Takizawa, Akiko Inoue, Akira Ishiko, Motonari Kondo, Yuriko Tanaka

**Affiliations:** 1Department of Molecular Immunology, Toho University School of Medicine, Tokyo 143-8540, Japan; md22011m@st.toho-u.jp (S.M.); cwwarita@med.toho-u.ac.jp (M.A.); kuwabara@med.toho-u.ac.jp (T.K.); taku.naitou@med.toho-u.ac.jp (T.N.); m20064t@st.toho-u.jp (S.T.); yurikota@med.toho-u.ac.jp (Y.T.); 2Department of Dermatology, Toho University Omori Medical Center, Tokyo 143-8541, Japan; akira.ishiko@med.toho-u.ac.jp; 3Graduate School of Medicine, Toho University, Tokyo 143-8540, Japan; 4Department of Otorhinolaryngology, Toho University Omori Medical Center, Tokyo 143-8541, Japan; akiko.inoue@med.toho-u.ac.jp

**Keywords:** autoimmunity, Sjögren’s disease, T cell hybridoma, T cell receptor, complementarity determining region 3

## Abstract

Sjögren’s disease (SjD), which is also known as Sjögren’s syndrome (SS), is a chronic autoimmune disease characterized by dysfunction of exocrine glands, such as the salivary and lacrimal glands, resulting in xerostomia (dry mouth) and keratoconjunctivitis sicca (dry eyes). Mice in which the SATB1 gene is conditionally deleted in hematopoietic cells (SATB1cKO mice) develop SS as early as 4 weeks of age; however, the etiology of the disease remains to be elucidated. Here, we found that the frequency of abnormally appearing CD4^+^CD8^+^ double positive (DP) T cells in the periphery of SATB1cKO mice was higher in the salivary glands than that in the spleen, suggesting a possible involvement of DP T cells in the pathogenesis of SS in SATB1cKO mice. To investigate the nature of DP T cells, we established DP T cell hybridomas by fusing T cells from the cervical lymph nodes of SATB1cKO mice with the BW5147 thymoma cell line. Among six DP hybridoma clones, the TCRβ gene from five clones exhibited a fetal or immature phenotype. In addition, four out of five clones exhibited upregulated transcription of IL-2 in the salivary glands of T/B cell-deficient RAG2^−/−^ mice, suggesting that autoreactive T cells were enriched in the DP T cell population of SATB1cKO mice. These results suggest that unusual DP T cells in SATB1cKO mice may be involved in autoimmune pathogenesis in SATB1cKO mice.

## 1. Introduction

Sjögren’s disease (SjD), which is also known as Sjögren’s syndrome (SS), is a chronic autoimmune disease characterized by dysfunction of systemic exocrine glands, including salivary and lacrimal glands [1,2]. As a result, patients with SS suffer from xerostomia (dry mouth) and keratoconjunctivitis sicca (dry eye) [3]. Although the etiology of SjD has not been clearly elucidated, humoral and/or cellular immunity are involved in tissue damage due to autoimmunity [4]. In humoral immunity, autoantibodies produced by self-reactive B cells may bind to self-organs and cause tissue damage in cooperation with complement [5]. T cells are the main players in cellular immunity and can recognize diverse antigen peptides. Generated self-reactive T cells during T cell development are eliminated by negative selection in the thymus [6]. However, self-reactive T cells do exist in the periphery, the activation of which is suppressed by peripheral tolerance mechanisms, in part by regulatory T (Treg) cells in healthy conditions [7]. Therefore, to understand the etiology of autoimmune diseases, it is important to uncover mechanisms by which tolerized autoreactive B and T cells are activated in the body. It is also necessary to clarify the nature of the pathogenic B and T cells that are involved in organ destruction.

Special AT-rich sequence binding protein-1 (SATB1) is a chromatin organizer that plays an important role in T cell development and functions [8,9]. In hematopoietic cell-specific SATB1 conditional knockout (SATB1cKO) mice, negative selection as well as positive selection during intrathymic T cell development are impaired [10]. As a result, the number of T cells is reduced in the periphery, and autoreactive T cells are not diminished efficiently in the thymus of SATB1cKO mice [10]. Furthermore, downregulation of CD4 or CD8 after positive selection at the CD4^+^CD8^+^ double positive (DP) stage is also impaired, resulting in the presence of peripheral DP T cells in SATB1cKO mice [10,11]. In addition, Treg cell development in the thymus is defective [12], which suggests a possible deficiency in immune tolerance in SATB1cKO mice. In fact, SATB1cKO mice develop SjD-like manifestations as early as 4 weeks of age [13,14]. The pathogenesis of SjD in SATB1cKO mice solely depends on T cells because symptoms characteristic of SjD, such as decreased saliva production, are observed in RAG2^−/−^ mice, which lack T and B cells, after transfer with T cells from the cervical lymph nodes (cLNs) derived from SATB1cKO mice [13]. Nevertheless, SATB1-deficient T cells do not respond to TCR stimulation very well, which is in part due to mitochondrial dysfunction because of a lack of mitochondrial transcription factor A (TFAM) expression [15,16]. Therefore, clarifying how T cells in SATB1cKO mice are involved in salivary gland destruction to gain further insights into the pathogenesis of T cell-dependent autoimmune diseases becomes worthwhile.

One T cell can recognize a single type of antigen peptide presented by major histocompatibility complex (MHC) molecules on the cell surface [17]. Therefore, to investigate the functions of antigen-specific T cells, T cell clones must be established in the presence of the T cell growth factor interleukin 2 (IL-2), with occasional antigen stimulation [18]. Alternatively, T cell hybridomas can be generated by fusing primary T cells with the thymoma cell line BW5147 [19]. A T cell hybridoma clone maintains the antigen-specificity of the primary T cells with infinite proliferative potential. As these T cell hybridomas can upregulate IL-2 production by stimulation via TCR, we can isolate T cell hybridoma clones specific for certain antigens of interest [20]. CD8^+^ T cell hybridomas are more difficult to establish than CD4^+^ T cell hybridomas because the CD8 gene is methylated and its expression is silenced in BW5147-based T cell hybridomas [13]. In this condition, since downregulation mechanisms of CD4 and CD8 expression in T cells are impaired in SATB1cKO mice [10,21], it might be possible to generate CD8^+^ and even DP T cell hybridomas by fusing BW5147 and SATB1-deficient T cells.

In this study, to obtain tools for the investigation into the nature of pathogenic T cells that cause SjD in SATB1cKO mice, we generated T cell hybridomas originating from T cells derived from the cLNs of SATB1cKO mice by fusion with BW5147 cells and obtained DP as well as CD4^+^ and CD8^+^ T cell hybridomas. We also found that the frequency of DP cell infiltration in damaged salivary glands is higher than that in the spleen of SATB1cKO mice. Therefore, we hypothesized that the DP T cell population would contain autoreactive T cells at a high frequency, and indeed, we could demonstrate that most DP hybridoma clones are autoreactive. In addition, based on the nucleotide sequence of the complementarity-determining region 3 (CDR3) region of the TCRβ genes in DP hybridoma clones, we discuss the possible origin of autoreactive DP T cells in SATB1cKO mice.

## 2. Results

### 2.1. DP T Cells Preferentially Locate in the Salivary Glands of SATB1cKO Mice That Develop SjD-like Manifestations

SATB1cKO mice are prone to autoimmunity and develop SjD-like manifestations, such as loss of saliva production and infiltration of immune cells in the salivary glands, as early as 4 weeks after birth [10,13]. Once saliva production decreases, massive T and B lymphocyte infiltration occurs in the salivary glands of SATB1cKO mice [13]. During the analysis of these infiltrated T cells, we found that the frequency of DP T cells in the salivary glands was higher than that in the spleens of SATB1cKO mice (Figure 1a). This high frequency of DP T cells was observed from the onset of the disease (4 weeks of age, Figure 1b) [13]. Although normality was not formally tested since the data did not show extreme skewness or outliers, parametric tests were also performed (Appendix A). Based on this observation, we hypothesized that the DP T cell population may contain autoreactive T cells in SATB1cKO mice. Therefore, we generated T cell hybridomas with T cells from SATB1cKO mice to investigate the nature of the DP T cells.

### 2.2. Successful Generation of DP T Cell Hybridomas from T Cells in the cLNs of SATB1cKO Mice

When we transferred SATB1-deficient T cells from the cLNs, draining lymph nodes of the salivary glands, into RAG2^−/−^ mice, we observed SjD-like symptoms in RAG2^−/−^ mice [13]. As SjD symptoms did not appear if T cells in the spleen of SATB1cKO mice were transferred to RAG2^−/−^ mice [13], we assumed that autoreactive T cells might be enriched in the cLNs. Therefore, we collected T cells from the cLNs of RAG2^−/−^ mice transferred with cLN T cells from SATB1cKO mice and generated T cell hybridomas by fusing T cells with BW5147. After cell fusion, hybridomas were grown in 78 wells. We examined the expression of TCR, CD4, and CD8 in cells from each well using FACS and found that DP as well as CD4^+^ and CD8^+^ T cell hybridomas were generated. The percentages of CD4^+^, CD8^+^, and DP T cell hybridomas were similar to the ratios of each T cell fraction in the cLNs of RAG2^−/−^ mice before cell fusion (Figure 2), suggesting that the cell fusion efficiency and initial expansion capability of hybridomas did not differ between CD4, CD8, and DP T cells.

### 2.3. Configuration of the TCR Genes in DP T Cell Hybridomas

To further analyze DP hybridomas, we first conducted limiting dilution to establish single clones and found that CD4 or CD8 expression was downregulated in some cases (Figure 3). Nevertheless, we could establish six DP T cell hybridoma clones (Figure 3, Table 1), although one clone (F7) was lost after RNA isolation. We cloned the complementary TCR genes from the total RNA derived from each line, sequenced them, and found that the TRBV12-1 Vβ segment was used in the TCRβ genes of three clones (Table 1). In the mice initiated with myelin oligodendrocyte glycoprotein (MOG)-induced experimental autoimmune encephalomyelitis (EAE), TRBV13-2 is preferentially used in the TCRβ gene of MOG-specific T cells [22]. Therefore, by FACS analyses, we examined Vβ usage in T cells in the cLNs of RAG2^−/−^ mice with SATB1cKO T cell transfer and found that the percentage of T cells with TRBV12-1 (Vβ5.2) Vβ segments detected by anti-Vβ5.1/Vβ5.2 antibodies was ~4% of the total (Appendix A). This result indicates that actual Vβ5.2 usage should be less than 4% in the T cell population before cell fusion. Therefore, it is unclear why three out of the six DP hybridoma clones were positive for a specific Vβ segment. In contrast to TCRβ, no preferential Vα usage was observed in the TCRα genes of the six DP hybridoma clones (Table 1), although proximal Vα segments were used in most clones, as reported previously [23]. It is worth noting that three clones expressed two different TCRα chains, which are sometimes observed in autoreactive T cells [24,25].

### 2.4. The CDR3 Regions of the TCRβ Genes in DP T Cell Hybridomas Exhibit Characteristics of a Fetal Type

Next, we examined the nucleotide sequences of the CDR3 regions of the TCRβ genes. During recombination between the D and J segments and V and D segments, additional sequences can be inserted at each junction (N region), which increases the diversity of the TCR pool [26,27]. In this gene rearrangement process, two different nucleotide insertions may occur. One is the palindrome (P) insertion, which is generated by the breaking of the hairpin formation at the end of the V, D, or J segments and DNA repair, resulting in a palindromic sequence to the terminal few nucleotides of the unadulterated coding joint [28,29]. The other is non-template (N) insertion, which can be caused by nucleotide misincorporation by DNA polymerase at the free ends during DNA repair and/or by terminal deoxynucleotidyl transferase (TdT), which catalyzes the template-independent addition of nucleotides [30,31]. As shown in Table 2, the insertion of a few (two or fewer) N nucleotides was observed in the CDR3 region of the TCRβ genes in all DP hybridoma clones, except for F7. As the expression of TdT in thymocytes appears 3–5 days after birth [26], the addition of a few N nucleotides to the N region of the TCRβ genes is characteristic of fetal or immature types of TCR [32]. Therefore, the majority of the DP cells used for the generation of T cell hybridomas may have been derived from the fetal (and/or newborn) thymus. In addition, the N region of the TCRβ genes seemed to be less heterogeneous because P nucleotide addition was also low (Table 2).

**Table 2 ijms-27-00414-t002:** CDR3 sequences of TCRβ chain from DP T cell hybridoma clones.

	CDR3 (β Chain)											Autoreactivity ^1^
Hybridoma	V Segment	V-D Junction			D Segment	D-J Junction			J Segment	Amino Acid Sequences	N Total	
		P	N	P		P	N	P				
F7	GCCAGCTCTCTC	GA	T		GGA		GCCG		ATTCGCCCCTCTAC	ASSLDGADSPLY	5	Lost
G5-5B	GCCAGCAGTCTTTC				AGGG	C	GG		GCTGAGCAGTTC	ASSLSGRAEQF	2	+
5	GCCAGCTCT	A	C		AG				CAAACACCGGGCAGCTCTAC	ASSTANTGQLY	1	+
35-10	GCCTGGAGTC			CC	GGGACAGGG				AACTCCGACTACACC	AWSPGTGNSDYT	0	−
37-11	GCCAGCTCTCAG				ACTGGGG		AG		GTGCAGAAACGCTGTAT	ASSQTGGGAETLY	2	+
C9-12-12	GCTAGCAGTAG				CTGGGG		AA		AAGACACCCAGTAC	ASSSWGKDTQY	2	+

^1^. IL-2 production was upregulated (+), or not (−), based on the results shown in Figure 4.

**Figure 4 ijms-27-00414-f004:**
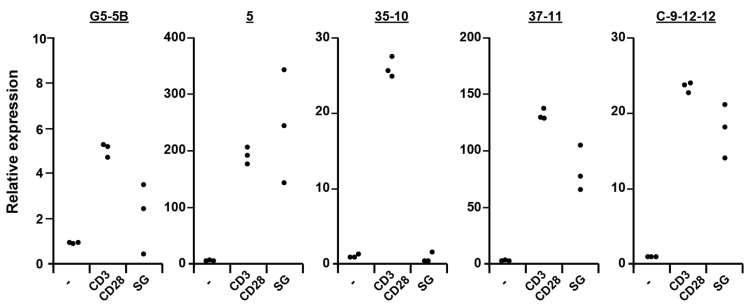
Identification of autoreactive DP hybridoma clones by injection into the salivary glands of RAG2^−/−^ mice (3 mice for each clone) and monitoring upregulation of IL-2 gene expression by quantitative PCR. Shown are the relative IL-2 gene expression levels over ZAP70 mRNA expression levels in each hybridoma clone without stimulation (-), with anti-CD3 and anti-CD28 antibodies stimulation (CD3/CD28), and harvested from the salivary glands 6 days after injection (SG) as described in the Section 4. The median value and interquartile range are indicated in Appendix A.

### 2.5. Most DP T Cell Hybridoma Clones Are Autoreactive

Finally, we examined whether DP T cell hybridomas could recognize autoantigens. When T cell hybridomas are stimulated with anti-CD3 and anti-CD28 antibodies, IL-2 expression is induced [19]. In fact, the transcription of the IL-2 gene was upregulated in all five DP hybridoma clones upon TCR stimulation, with some clonal variation in the expression levels of IL-2 transcripts (Figure 4). To investigate whether any of these DP T cell hybridoma clones were autoreactive, we injected each DP hybridoma clone directly into the salivary glands of RAG2^−/−^ mice, removed the glands 6 days after injection, and harvested the hybridoma cells as described in the Section 4. Following the extraction of total RNA from the recovered hybridomas, we examined the transcript levels of the IL-2 gene and found that four out of five clones (except clone 35–10) were autoreactive (Figure 4, Table 2). Therefore, although it is unclear whether such autoreactive DP T cells are truly pathogenic T cells in this study, we concluded that DP T cells in the salivary glands of SATB1cKO mice were autoreactive at high frequencies.

## 3. Discussion

T cells recognize complexes with single-antigenic peptides and self-MHCs via highly diversified TCR [33]. As one T cell expresses one type of TCR, with some exceptions such as some T cells expressing two different TCRα chains [24,25], antigen specificity of T cells needs to be investigated at the clonal level. Therefore, T cell hybridomas are useful tools in the research of T cells [19,34]. In this study, we generated T cell hybridomas by fusing T cells from SATB1cKO mice with the thymoma cell line BW5147. Importantly, we established six DP T cell hybridoma clones, isolated TCR genes from each clone, and stably maintained five clones. As DP T cells are not usually present in the periphery, to our knowledge, this is the first demonstration of the establishment of DP T cell hybridomas from peripheral T cells.

To date, the generation of CD8^+^ T cell hybridomas is considered more difficult than that of CD4^+^ T cell hybridomas because the CD8 gene locus is methylated and CD8 expression is downregulated in hybridomas generated by cell fusion with BW5147 and CD8^+^ T cells [35]. Nevertheless, we could generate CD4^+^, CD8^+^, and DP T cell hybridomas in this study, although we did not confirm the establishment of stable CD8^+^ T cell hybridoma clones. Yet, we preliminarily analyzed the expression of SATB1 in BW5147 cells and found that its expression levels were low, suggesting that the successful generation of CD8^+^ and DP T cell hybridomas by fusing BW5147 and SATB1cKO T cells may be related to SATB1 deficiency.

The frequency of DP T cells in the salivary glands, where autoimmune organ destruction occurs, was higher in SATB1cKO mice than that in the spleen (Figure 1), suggesting that pathogenic T cells that lead to SjD in SATB1cKO mice may be enriched in the DP T cell population. We analyzed the TCR genes in six DP hybridoma clones and found that the TCRβ genes from five clones had two or fewer N nucleotides inserted in the N region of CDR3, which resembled T cells in TdT-deficient mice, as well as T cells developed in fetuses [26,30,31,32]. Therefore, most DP T cells may originate at the fetal stage.

In this study, we developed a novel method to identify autoreactive, perhaps salivary gland-specific autoantigen-reactive T cell hybridomas by injecting cells directly into the salivary glands. Although it is unclear whether any of the DP T cell hybridoma clones were truly pathogenic, four of the five clones tested were autoreactive in the salivary glands (Figure 4, Table 2). Previously, we demonstrated that Treg cells were absent from infant SATB1cKO mice until 1 week after birth [13]. If Treg cells are injected into SATB1cKO mice at 3 days of age, the development of SjD is largely suppressed, implying that activated T cells in the absence of Treg cells may be pathogenic T cells in SATB1cKO mice [13]. As the addition of N nucleotides is mostly mediated by TdT, whose expression is initiated on day 3–5 after birth in thymocytes, it is reasonable to assume that pathogenic T cells for SjD that are activated within 1 week after birth, when Treg cells are absent, exhibit fetal types [26,30,31,32]. To gain more insight into the nature of pathogenic T cells in SATB1cKO mice, we need to comprehensively analyze not only DP T cells but also CD4^+^ and CD8^+^ T cells using single-cell RNA-sequencing techniques, for example. We also need to clarify whether DP T cells receive antigen presentation via MHC class I or II molecules. Further studies are necessary to fully understand how SjD develops in SATB1cKO mice and to find a way to reduce or prevent SjD symptoms in these mice.

## 4. Materials and Methods

### 4.1. Mice

SATB1cKO (SATB1^fl/fl^Vav-Cre^+^) mice were generated as previously described [10,13]. RAG2^−/−^ mice were maintained in our laboratory. The C57BL/6 mice were obtained from CLEA Japan (Tokyo, Japan). All mice used in this study had a C57BL/6 background and were maintained under specific pathogen-free conditions at an animal facility at the Toho University School of Medicine. All experiments involving mice were approved by the Toho University Administrative Panel for Animal Care (24-581) and recombinant DNA use (24-587).

### 4.2. Flow Cytometry

The following antibodies were used for cell surface staining: CD4 (GK1.5) PE-Cy7, CD8 (53–6.7) Pacific Blue, and TCRβ APC (TONBO Biosciences, San Diego, CA, USA). Thy1.2 (30-H12) PE was purchased from eBioscience (San Diego, CA, USA). The mouse Vβ TCR screening panel (FITC) was purchased from BD Biosciences (San Jose, CA, USA). Cells from the spleen or cLNs were depleted of erythrocytes and stained with fluorophore-conjugated antibodies. Samples with 7-amino-actinomycin D (7-AAD, BD Biosciences, Mountain View, CA, USA) were analyzed or sorted using FACSCanto II, FACSFortessa, or FACSAria Fusion flow cytometers (BD Biosciences), and the data were analyzed using FlowJo software (version 9.8.1, Tree Star, Ashland, OR, USA). In brief, after dead cells and doublets were excluded, we gated on the TCRβ^+^ or Thy1.2^+^ T cell population and analyzed CD4, CD8, or Vβ5.1/5.2 expression.

### 4.3. Generation of T Cell Hybridomas

Thy1.2^+^ cells were sorted from the cLNs of SATB1cKO mice (12–15 weeks old) and transferred into RAG2^−/−^ mice (8–12 weeks old) via intravenous injection. Cells were harvested from the cLNs of RAG2^−/−^ mice at 12 weeks after T cell transfer and mixed with BW5147 cells at 1:1 ratio in pre-warmed serum-free RPMI 1640 medium (Invitrogen, Carlsbad, CA, USA). After centrifuging, the supernatant was aspirated, and we added pre-warmed 1 mL 50% polyethylene glycol (PEG; NACALAI TESQUE, INC., Kyoto, Japan), which was slowly added for one minute for cell fusion. While maintaining the 50 mL conical centrifuge tube at 37 °C, serum-free RPMI 1640 medium was added to dilute the PEG in two steps: 2 mL over 2 min, followed by 7 mL over 2 min. After adding an additional 30 mL of pre-warmed serum-free RPMI 1640 medium, the cells were centrifuged, and the supernatant was removed. The pellet was then resuspended in culture medium containing 10% FCS to a final concentration of 1 × 10^6^ cells/mL, and 0.1 mL of the cell suspension was dispensed into each well of a 96-well plate (Corning, Glendale, AZ, USA). After 24 h, 100 µL of culture medium containing 10% FCS and a double concentration of HAT supplement (Thermo Fisher Scientific, Walthaw, MA, USA) was added to each well. HAT-resistant clones were selected approximately 3 weeks later, and the expression of CD4, CD8, and TCRβ was subsequently analyzed by FACS.

### 4.4. TCR Gene Cloning

To extract total RNA using Isogen II (Nippon Gene, Tokyo, Japan), 1–3 × 10^6^ DP hybridoma cells were used. First-strand cDNA was synthesized from 1 µg of total RNA with SuperScript III reverse transcriptase (Thermo Fisher Scientific) and oligo(dT)30 primer. The primer was pre-annealed to mRNA with dNTPs, and the cDNA strand was synthesized in a 20 µL reaction containing DTT, RNase inhibitor, and reverse transcriptase. One microliter of cDNA was used to amplify the coding regions of TCR genes expressed in the hybridoma clones by nested PCR using PrimeStar Max DNA polymerase (Takara Bio, Shiga, Japan). First, PCR was performed in a 20 µL reaction with the V segment-specific forward primers mixture described by Hamana et al. [15] (Appendix A) and C segment UTR-specific reverse primer (Appendix A). Mixtures of mAL and Trac 3UTR PrR1 primers for the α chain and mBL and Trbc1 or Trbc2 3UTR PrR1 primers for the β chain were used. The second PCR was performed using a tenth of the first PCR as a template and primers designed for seamless assembly with a pCR2.1 vector (Appendix A). The coding regions of TCR α and β chains in the second PCR were amplified with a pair of P2A-C IF and Trac 3UTR IF R2 primers as well as a pair of BES-AP IF and Trbc1 or Trbc2 3UTR IF R2 primers, respectively. The PCR products were cloned by seamless assembly into a pCR2.1 fragment using the In-Fusion HD cloning kit (Takara Bio), which was prepared by inverse PCR with PrimeStar Max DNA polymerase and the primer pair of pCR2.1F and pCR2.1R. Cloned TCR chains were sequenced using primers M13 rev or T7pro (Appendix A).

### 4.5. Examination of Autoreactivity of T Cell Hybridoma Clones

Into the right and left submandibular glands of RAG2^−/−^ mice under anesthesia with a mixture of 0.75 mg/kg medetomidine (Nippon Zenyaku Kogyo, Koriyama, Japan), 4 mg/kg midazolam (SANDOZ, Yamagata, Japan), and 5 mg/kg butorphanol tartrate (Meiji Seika Pharma, Tokyo, Japan), 1 × 10^6^ hybridoma cells were injected. At 6 days after injection, both submandibular glands were removed and cut in HBSS containing 5% FCS. Collagenase A (1 mg/mL; Roche Diagnostics, Mannheim, Germany), DNase I (0.2 mg/mL; Sigma-Aldrich, St. Louis, MO, USA), and 10 mM HEPES were then added, and the tissues were incubated at 37 °C in 5% CO_2_ for 35 min. After incubation, 20 mM EDTA was added, and the cells were washed and resuspended in 40% Percoll Plus (Merck KGaA, Darmstadt, Germany), which was overlaid onto 80% Percoll Plus. The samples were centrifuged at 2400 rpm for 20 min at 25 °C with both acceleration and braking turned off. Cells collected from the intermediate phase were used for total RNA extraction with Isogen II. Quantitative PCR was performed using the TaqMan Gene Expression Assay kit (Thermo Fisher Scientific) according to the manufacturer’s instructions on an ABI 7500 Fast system (Thermo Fisher Scientific). The following primers (Thermo Fisher Scientific) were used: Hprt, Mm00446968_m1; Il2, Mm00434256_m1; and Zap70, Mm00494260_m1. IL-2 mRNA expression levels were normalized over ZAP70 mRNA levels. With this procedure, although IL-2-producing hybridomas in the salivary glands are determined as auto-reactive clones, they cannot be conclusively defined as salivary gland antigen-specific clones.

### 4.6. Statistical Analysis

Statistical analysis was performed using Student’s *t*-test to compare means, the Mann–Whitney U test, or the Wilcoxon signed-rank test with the assumption of unequal variance and a confidence level of 95%. Intragroup comparison was made in the analysis shown in Figure 1b.

## Figures and Tables

**Figure 1 ijms-27-00414-f001:**
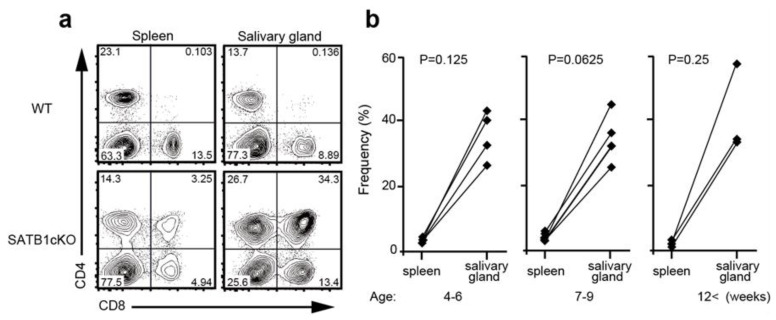
DP T cells are preferentially localized in the salivary glands of SATB1cKO mice. (**a**) Expression of CD4 and CD8 in the cells derived from the spleen (left) or infiltrated cells in the salivary glands (right) of wild-type (WT) (upper panels) and SATB1cKO (lower panels) mice at 8 weeks of age. (**b**) Frequency of DP T cells in the total (CD4^+^, CD8^+^, and DP) T cells in the spleen and salivary glands of SATB1cKO mice at different ages indicated. At each time point, 4 mice (4–6 weeks old), 5 mice (7–9 weeks old), and 3 mice (over 12 weeks old) were used. Intragroup comparison by Wilcoxon signed-rank test was made. Median (IQR) value of DP T cell percentage in the spleen and the salivary gland was 3.21 (2.75–3.79) vs. 36.6 (28.2–42.6) at 4–6 weeks old; 3.01 (2.94–4.22) vs. 32.1 (28.8–40.5) at 7–9 weeks old; and 1.87 (1.17–3.25) vs. 34.3 (33.5–57.8) at over 12 weeks old.

**Figure 2 ijms-27-00414-f002:**
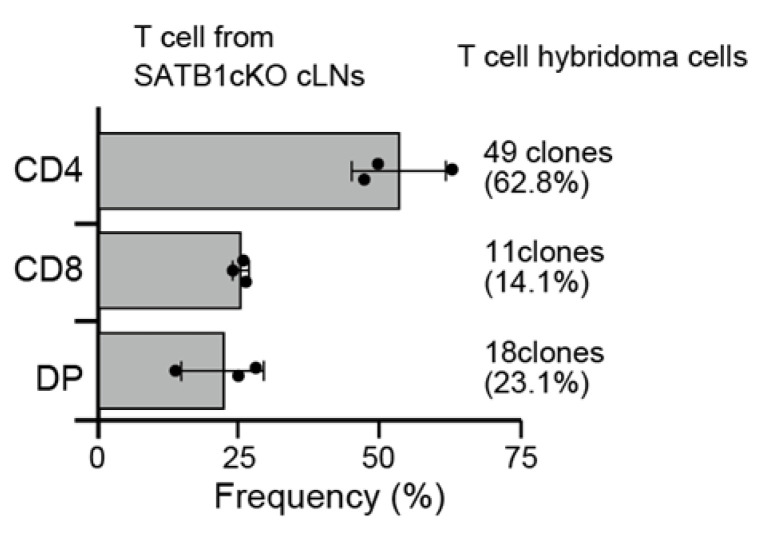
Percentage of CD4^+^, CD8^+^, and DP T cells in the cLNs of RAG2^−/−^ mice transferred with SATB1cKO cLN T cells before cell fusion (*n* = 3). The mean value is indicated by bars. The number of initially obtained hybridoma types, with the percentage in parenthesis, is also shown.

**Figure 3 ijms-27-00414-f003:**
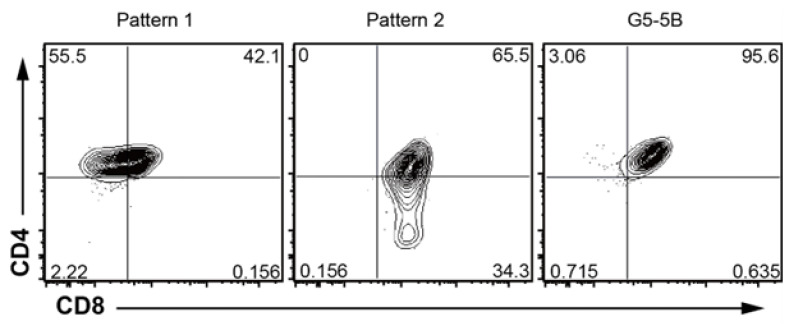
CD4 and CD8 expression is downregulated in some DP hybridoma clones during culture. After limiting dilution, the expression of either CD8 (pattern 1, 5 clones out of 6 established clones) or CD4 (pattern 2, 1 clone) was downregulated during establishment of DP clones. In contrast, established DP clones stably expressed both CD4 and CD8 molecules on their cell surfaces (right panel as an example).

**Table 1 ijms-27-00414-t001:** TCR repertoire and autoreactivity of DP T cell hybridoma.

Hybridoma	α Chain			β Chain			
	V Segment	J Segment	C Segment	V Segment	D Segment	J Segment	C Segment
F7	TRAV13-4/TRDV 7	TRAJ26	TRAC	TRBV12-1	TRBD1	TRBJ1-6	TRBC1
	TRAV3-4	TRAJ56	TRAC				
G5-5B	TRAV21/TRDV12	TRAJ53	TRAC	TRBV26	TRBD1	TRBJ2-1	TRBC2
5	TRAV16	TRAJ58	TRAC	TRBV12-1	TRBD1	TRBJ2-2	TRBC2
35-10	TRAV7-3	TRAJ49	TRAC	TRBV31	TRBD1	TRBJ1-2	TRBC1
	TRAV6-7/TRDV9	TRAJ48	TRAC				
37-11	TRAV7D-3	TRAJ50	TRAC	TRBV12-1	TRBD2	TRBJ2-3	TRBC2
	TRAV6-6	TRAJ48	TRAC				
C-9-12-12	TRAV4-4/TRDV10	TRAJ53	TRAC	TRBV17	TRBD2	TRBJ2-5	TRBC2

## Data Availability

The original contributions presented in this study are included in the article/Appendix A. Further inquiries can be directed to the corresponding author.

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
