# Peer review of "Int. J. Mol. Sci.2026, 27(1), 414;https://doi.org/10.3390/ijms27010414"

_ijms, 2025, doi:10.3390/ijms27010414_

Round 1

Reviewer 1 Report

Comments and Suggestions for Authors

In their study, Mashima et al. raised a relevant and interesting topic in modern rheumatology: the role of DP T cells in the pathogenesis of Sjögren's syndrome. The study yielded DP T cell hybridomas demonstrating autoreactive properties. The manuscript appears extremely interesting both from the methodological standpoint of hybridoma creation and from the results obtained. However, many of the data presented in the figures in the "Results" section are not clearly presented in the text of the manuscript itself, which requires correction. Furthermore, the statistical approach to data calculation should be more fully described, as well as the number of subjects included in the compared samples. Moreover, the lack of a specific study objective is noteworthy. The proposed changes will improve the quality of the manuscript.

Suggestions:

  1. Line 18. Please replace "CD4+CD8+ double-positive (DP) cells" with "CD4+CD8+ double-positive (DP) T cells."
  2. Lines 28–29. Please remove the numbers after your keywords.
  3. In the "Introduction" section, please describe the specific objective of the study.
  4. Lines 80–86. Please provide the median and interquartile range of the obtained data (DP T cell concentrations) in the salivary glands and spleens of both groups of mice. Furthermore, the tests used for intra- and intergroup comparisons must be specified in the Materials and Methods.
  5. Lines 88–93. Was a test for normality of data distribution performed? The number of mice in both groups must be indicated.
  6. Lines 96–108. Please provide specific numbers for CD4+ T cell, CD8+ T cell, and DP T cell concentrations in the text. It is best to present the data as medians and interquartile ranges. Also, provide p-values for comparisons of total T cell concentrations between the "T cells from SATB1cKO cLNs" and "T cell hybridoma cells" groups in the text.
  7. Lines 110–112. Figure 2 is best presented as scatter plots to clearly show the distribution and number of points in both groups. In the figure legend, indicate the number of wells studied in both groups.
  8. Line 122. Present the obtained data in the text of the manuscript or in a supplement files.
  9. Lines 129–132. Indicate the number of clones (n) that exhibit the patterns shown in the figure.
  10. Line 133. Why are some sections of the table highlighted in yellow color?
  11. Line 154. Why are some sections of the table highlighted in yellow color?
  12. Lines 160–163. It is necessary to indicate whether the hybridoma-derived DP T cells were specific to salivary gland cell antigens. Was the autoreactivity of these DP cells to antigens from cells of other tissues studied? If not, this should be added to the limitations in the "Materials and Methods" section.
  13. Line 169. IL-2 gene expression should be indicated in Figure 4.
  14. Lines 171–172. It is unclear whether the data presented are for IL-2 gene expression after administration of DP T cells into the salivary glands of RAG2-/- mice or for IL-2 gene expression after stimulation with CD3/CD28 antibodies. The wording in the figure legend should be revised. Furthermore, IL-2 gene expression should be compared between the hybridoma groups and the median values with interquartile range, as well as the p-value, should be provided in the text of the manuscript. The p-value should also be provided in Figure 4. Figure 4 is best presented as scatter plots to clearly understand the distribution and number of points in groups.
  15. Please, add the gating strategy of lymphocytes in the "Materials and Methods" section.

Author Response

Reviewer 1

In their study, Mashima et al. raised a relevant and interesting topic in modern rheumatology: the role of DP T cells in the pathogenesis of Sjögren's syndrome. The study yielded DP T cell hybridomas demonstrating autoreactive properties. The manuscript appears extremely interesting both from the methodological standpoint of hybridoma creation and from the results obtained.

Reply: It is our great plesuare that this reviewer found scientific significance of our paper. We really appreciate it.

However, many of the data presented in the figures in the "Results" section are not clearly presented in the text of the manuscript itself, which requires correction.

Reply: We edited the whole text, especially the Results section, so that readers of this paper could understand the significance of this paper better.

Furthermore, the statistical approach to data calculation should be more fully described, as well as the number of subjects included in the compared samples.

Reply: Based on the comments of this reviewer, we re-analyzed the data and modified the figures as we described below.

Moreover, the lack of a specific study objective is noteworthy.

Reply: We modified the Introduction section so that the purpose of this study would be clearer.

The proposed changes will improve the quality of the manuscript.

Reply: Certainly, all comments are helpful to improve this paper.

Specific comments

#1: Please replace "CD4+CD8+ double-positive (DP) cells" with "CD4+CD8+ double-positive (DP) T cells."

#10: Why are some sections of the table highlighted in yellow color?

#11: Why are some sections of the table highlighted in yellow color?

Reply: We checked the Word file of this paper and found no errors at the parts pointed out by the reviewer. This might be caused by an incompatibility issue between different versions of the OS or other software. We will carefully check the gally proof during the final publication process.

#2: Please remove the numbers after your keywords.

Reply: We removed the numbers in the Keywords section.

#3: In the "Introduction" section, please describe the specific objective of the study.

Reply: We modified a sentence in the Introduction section (lines 72-73).

#4: Please provide the median and interquartile range of the obtained data (DP T cell concentrations) in the salivary glands and spleens of both groups of mice. Furthermore, the tests used for intra- and intergroup comparisons must be specified in the Materials and Methods.

#5: Lines 88–93. Was a test for normality of data distribution performed? The number of mice in both groups must be indicated.

Reply: The reviewer asked us how we analyzed the data shown in Figure 1b. We performed intragroup comparisons between the salivary glands and spleens with the numbers of mice, now indicated in the figure legends for Figure 1b (lines 95-96). To clearly present the results of the intragroup comparisons, we modified Figure 1b, as shown in the manuscript (lines 96-97). We indicated the median and interquartile range in the text (lines 97-99) in this revised version. Given the robustness of the t-test to deviations from normality, especially with small sample sizes in this case, a formal normality test was not performed. In this revised version, the original Figure 1b is shown as Figure S1 as indicative information. We added the explanation in the Materials and Methods section as reviewer suggested (lines 282-284).

#6: Please provide specific numbers for CD4+ T cell, CD8+ T cell, and DP T cell concentrations in the text. It is best to present the data as medians and interquartile ranges. Also, provide p-values for comparisons of total T cell concentrations between the "T cells from SATB1cKO cLNs" and "T cell hybridoma cells" groups in the text.

#7: Figure 2 is best presented as scatter plots to clearly show the distribution and number of points in both groups. In the figure legend, indicate the number of wells studied in both groups.

Reply: We are sorry for causing confusion due to the presentation of the results shown in Figure 2. To clearly show the fact that the frequency of CD4+, CD8+, and DP T cells before and after cell fusion was similar in this study, we modified Figure 2 in its current form. Now, the reviewer may find that a statistical comparison cannot be performed in this case.

#8: Present the obtained data in the text of the manuscript or in a supplement files.

Reply: We added a FACS plot that shows Vb5.1/ Vb5.2 expression on T cells in the cLNs of RAG2-/- mice with SATB1cKO T cell transfer to the supplemental materials (Figure S2).

#9: Indicate the number of clones (n) that exhibit the patterns shown in the figure.

Reply: Among 6 established DP clones, 5 and 1 clones exhibited patterns 1 and 2, respectively. We modified the figure legends for Figure 3 (lines141-142).

#12: It is necessary to indicate whether the hybridoma-derived DP T cells were specific to salivary gland cell antigens. Was the autoreactivity of these DP cells to antigens from cells of other tissues studied? If not, this should be added to the limitations in the "Materials and Methods" section.

Reply: The reviewer pointed a very important issue. We understand that with our assays in this study, we cannot elucidate DP T cell hybridomas specific to salivary gland-specific antigens. We underlined the sentences regarding this issue (lines 175-177, lines 213-217). We also added sentences in the Materials and Methods section (lines 278-280) as the reviewer suggested.

#13: IL-2 gene expression should be indicated in Figure 4.

#14: Lines 171–172. It is unclear whether the data presented are for IL-2 gene expression after administration of DP T cells into the salivary glands of RAG2-/- mice or for IL-2 gene expression after stimulation with CD3/CD28 antibodies. The wording in the figure legend should be revised. Furthermore, IL-2 gene expression should be compared between the hybridoma groups and the median values with interquartile range, as well as the p-value, should be provided in the text of the manuscript. The p-value should also be provided in Figure 4. Figure 4 is best presented as scatter plots to clearly understand the distribution and number of points in groups.

Reply: Again, we are sorry for this confusion. Figure 4 shows the IL-2 expression levels of 5 DP hybridoma clones. We first examined the IL-2 expression levels of each clone upon CD3/CD28 stimulation in vitro, which can be considered as the fully stimulated condition. Now, we displayed the results with scatter plots. The median value and interquartile range are indicated in Table S1.

#15: Please, add the gating strategy of lymphocytes in the "Materials and Methods" section.

Reply: We added the gating strategy in the Materials and Methods section (lines 228-231).

Reviewer 2 Report

Comments and Suggestions for Authors

In this article, the authors aimed to reveal the pathogenetic features of autoimmune process in SATB1cKO mouse model of SS. Interesting observations have been made regarding increased frequency of CD4+CD8+ DP T cells in salivary glands of SATB1cKO mice. To gain further understanding of nature of these cells, DP T cell hybridomas have been established, and subsequent sequencing of TCR chains revealed an immature phenotype of CDR3 regions of TCRb. This work contributes to the understanding of the SS pathogenesis in SATB1cKO mice. However, it may benefit from addressing some points.

Some general concerns:

According to last recommendations https://www.nature.com/articles/s41584-025-01268-z, the term Sjögren’s syndrome is officially shifting to Sjögren's Disease, because it indicates the serious and systemic nature of the disease. To reflect the updated terminology, authors could replace the term and abbreviation from SS to SjD.

Phenotype of T cells should be mentioned (in Introduction)

Other concerns:

Line 32-33: the statement about salivary and lacrimal gland dysfunction could be expanded with indication that all exocrine glands are involved in pathogenesis. I think this will highlight the systemic nature of the disease.

Lines 35-37: as the article makes a contribution to the cellular aspect of pathogenesis, it seems reasonable to expand this section and describe in more details involved in the autoimmune response cell types, their interactions and functioning.

Moreover, statement that «T cells are the primary cause of autoimmune diseases» requires correction, as autoimmunity is a multifactorial pathology with a lot of factors acting such as «triggers», which together provoke uncontrolled T cell activation. T cells itself do not act as a cause, but are the primary target for regulatory/therapeutic intervention.

Lines 38-39: I would advise to describe more precisely the relationship between SATB1 and T cell functions. Maybe it would also be worthwhile to draw a parallel with the functions of SATB1 in human T cells?

Lines 40-41: It would be helpful for the reader to place a more detailed information regarding impaired positive and negative selection of T cells in SATB1cKO mice. Lines 56-57 contain related information about impaired CD4 and CD8 downregulation which could be placed to the line 41 to maintain a logical sequence.

This paragraph lacks information on T cell phenotype.

Paragraph 2.3: This part could benefit from adding information on what is known about the involvement of specific V segments of TCRb in autoimmunity.

It may also be interesting and helpful for future research to discuss (maybe in Discussion) what may influence the stability of CD4 and CD8 expression in DP hybridomas.

Table 1: typo in the title («DO T cell hybridoma» instead of DP). Also, the last 3 columns (N total, CDR3 amino acid numbers and IL-2 production) should be removed from this table, because this data is discussed later in the text.

«N total» and «CDR3 amino acid numbers» fit better to the Table 2, because in paragraph 2.4 N insertions in CDR3 region are discussed in details. It is also recommendable to add a figure or a scheme of the VDJ recombination processes and nucleotide insertions types, pointing to the possible sites of insertion.

CDR3 amino acid length could be also analysed and discussed, as short length of CDR3 increases the potential for self-recognition and makes TCR more prone to autoimmunity.

Paragraph 2.5:

It could be interesting to discuss, why one hybridoma clone (35-10) was not considered auto-reactive. Can it be possible that there is a relationship with the «most» immature phenotype of CDR3b among all hybridomas analyzed with N insertions = 0?

Moreover, as studies report polyfunctionality of DP T cells, it is worthwhile to study functional activity of these cells in the future.

It seems also interesting to study which factors contribute to the maintenance of this DP T cell phenotype and functioning in infiltration sites.

Some references, for example those on the nucleotide insertions during VDJ recombination, could be updated to reflect the current state of the knowledge in the field.

I recommend major revision before publication.

Author Response

Reviewer 2

General concerns

According to last recommendations https://www.nature.com/articles/s41584-025-01268-z, the term Sjögren’s syndrome is officially shifting to Sjögren's Disease, because it indicates the serious and systemic nature of the disease. To reflect the updated terminology, authors could replace the term and abbreviation from SS to SjD.

Reply: As we mentioned above, we changed the term Sjögren’s syndrome (SS) to Sjögren's disease (SjD) in this paper based on this suggestion.

Phenotype of T cells should be mentioned (in Introduction)

Reply: We added the characteristics of SATB1-deficient T cells in the Introduction section (lines 50-51, 56-58).

Other concerns

Line 32-33: the statement about salivary and lacrimal gland dysfunction could be expanded with indication that all exocrine glands are involved in pathogenesis. I think this will highlight the systemic nature of the disease.

Reply: We refraised the sentence so that readers could find that any exocrine glands in the body can be targets of autoimmunity in SjD patients (lines 14-15, 34-35).

Lines 35-37: as the article makes a contribution to the cellular aspect of pathogenesis, it seems reasonable to expand this section and describe in more details involved in the autoimmune response cell types, their interactions and functioning.

Reply: We modified the part pointed out by the reviewer to better clarify fundamental questions of autoimmune diseases (lines 40-45).

Moreover, statement that «T cells are the primary cause of autoimmune diseases» requires correction, as autoimmunity is a multifactorial pathology with a lot of factors acting such as «triggers», which together provoke uncontrolled T cell activation. T cells itself do not act as a cause, but are the primary target for regulatory/therapeutic intervention.

Reply: By modifying the paragraph, we removed this sentence in the revised version.

Lines 38-39: I would advise to describe more precisely the relationship between SATB1 and T cell functions. Maybe it would also be worthwhile to draw a parallel with the functions of SATB1 in human T cells?

Response: As mentioned above, we briefly described the characteristics of SATB1-deficient T cells in the Introduction section (lines 46-58). The functions of SATB1 in human T cells have not been documented very well. Therefore, we did not add this issue to the revised version of this paper.

It would be helpful for the reader to place a more detailed information regarding impaired positive and negative selection of T cells in SATB1cKO mice. Lines 56-57 contain related information about impaired CD4 and CD8 downregulation which could be placed to the line 41 to maintain a logical sequence.

Response: We have explained the phenotype of SATB1cKO mice caused by impairment of positive and negative selections published in our previous papers. We also moved the explanation of impairment of CD4 and CD8 downregulation after positive selection in SATB1cKO mice. Please see lines 50-51.

This paragraph lacks information on T cell phenotype.

Response: Although it is not clear which paragraph the reviewer is referring to, we edited the whole text to fix any insufficient descriptions.

Paragraph 2.3: This part could benefit from adding information on what is known about the involvement of specific V segments of TCRb in autoimmunity.

Response: In MOG-induced EAE mice, TRBV13-2 is preferentially used in the TCR of MOG-specific T cells. We added this issue to lines127-129.

It may also be interesting and helpful for future research to discuss (maybe in Discussion) what may influence the stability of CD4 and CD8 expression in DP hybridomas.

Response: We agree with the reviewer. We briefly described this issue in the Discussion section (lines 192-198, underlined). In our preliminary experiments, we examined the expression of Thpok and Runx3 in DP hybridomas. But, we could not obtain any conclusive results so far. Since this result is too preliminary, we would not like to mention this in this paper.

Table 1: typo in the title («DO T cell hybridoma» instead of DP). Also, the last 3 columns (N total, CDR3 amino acid numbers and IL-2 production) should be removed from this table, because this data is discussed later in the text.

Reply: We corrected all typos and modified the tables as the reviewer suggested.

«N total» and «CDR3 amino acid numbers» fit better to the Table 2, because in paragraph 2.4 N insertions in CDR3 region are discussed in details. It is also recommendable to add a figure or a scheme of the VDJ recombination processes and nucleotide insertions types, pointing to the possible sites of insertion.

Reply: Based on this suggestion, we modified the tables as we described above. Since this is not a review article, we did not add any schematic explanations of the VDJ recombination processes. We would like readers to check the references we referred in this paper, if necessary. Although Ref 28 is old, it is still a good review for understanding the formation of the N region during VDJ recombination processes. As the reviewer asked us to add updated the references, we added Ref 29, although this was published in 2004.

It could be interesting to discuss, why one hybridoma clone (35-10) was not considered auto-reactive. Can it be possible that there is a relationship with the «most» immature phenotype of CDR3b among all hybridomas analyzed with N insertions = 0?

Reply: Certainly, it is interesting to know the reason why 35-10 clone is not autoreactive. Probably, a more interesting point to discuss would be why the other 4 clones are autoreactive. As we described in the Discussion section (lines 213-217, underlined), more comprehensive studies are necessary in the future to gain more insights into the nature of pathogenic T cells in SATB1cKO mice.

It seems also interesting to study which factors contribute to the maintenance of this DP T cell phenotype and functioning in infiltration sites.

Moreover, as studies report polyfunctionality of DP T cells, it is worthwhile to study functional activity of these cells in the future.

Reply: In our previous studies, we found that IDO expression is upregulated in affected salivary glands (Ref 14). In our preliminary examination, some DP T cells produce IFNg (unpublished observations). But, many studies need to be performed again to uncover the etiology of SjD in SATB1cKO mice. We really appreciate Reviewer 2's interest in our study.

Some references, for example those on the nucleotide insertions during VDJ recombination, could be updated to reflect the current state of the knowledge in the field.

Response: We tried to add references that are as recent as possible. But, in the case of VDJ recombination, for example, we could not find any appropriate reviews regarding generation processes of the CDR3 region during gene recombination processes. Nevertheless, we tried to do our best.

Round 2

Reviewer 1 Report

Comments and Suggestions for Authors

Mashima et al. have done a great deal of serious work to improve the quality of the manuscript. The study is now presented comprehensively enough for easy comprehension by readers. The manuscript is of high quality, but contains several inaccuracies. In the supplementary file, Figure S1 and Table S1 are inconsistent with their roles in the text of the manuscript. Furthermore, Figure S2 is missing from the supplementary file. It is possible that the authors accidentally uploaded an outdated version of the supplementary file. Nevertheless, after correcting these inaccuracies, I can recommend the manuscript for publication.

Author Response

Because of the issue of online submission systems, we could not upload the PDF file with supplemental materials at the revision.  This is the reason why Reviewer 1 saw the original, not revised supplemental materials. Now we attached supplemental materials at the end of the Word text. I hope Reviewer 1 will find that we have sufficiently revised the manuscript.

Reviewer 2 Report

Comments and Suggestions for Authors

The authors addressed all my initial concerns. I recommend to accept manuscript for publication

Author Response

Thank you for reviewing our paper. We really appreciate it.